# A Plaque Instability Index Calculated by Histological Marker Analysis of the Endarterectomy Carotid Artery

**Doina Butcovan [1,2], Veronica Mocanu [3,*] , Mihai Enache [4,5], Beatrice Gabriela Ioan [6,7] and Grigore Tinica [4,5]**

1 Department of Morpho-Functional Sciences (Morphopathology), "Grigore T. Popa" University of Medicine and Pharmacy, 16, Universitatii Street, 700115 Iasi, Romania
2 Department of Pathology, "Prof. George Georgescu" Institute of Cardiovascular Diseases, 50, Carol I Avenue, 700503 Iasi, Romania
3 Department of Morpho-Functional Sciences (Pathophysiology), "Grigore T. Popa" University of Medicine and Pharmacy, 16, Universitatii Street, 700115 Iasi, Romania
4 Department of Cardiovascular Surgery, "Prof. George Georgescu" Institute of Cardiovascular Diseases, 50, Carol I Avenue, 700503 Iasi, Romania
5 Department of Surgery (Cardiac Surgery), "Grigore T. Popa" University of Medicine and Pharmacy, 16, Universitatii Street, 700115 Iasi, Romania
6 Institute of Forensic Medicine, 4 Buna Vestire Street, 700455 Iasi, Romania
7 Department of Internal Medicine (Legal Medicine), "Grigore T. Popa" University of Medicine and Pharmacy, 16, Universitatii Street, 700115 Iasi, Romania
* Correspondence: veronica.mocanu@umfiasi.ro

**Abstract:** (1) Background: Atherosclerosis is a pandemic condition that causes the development of ischemic syndromes including myocardial infarctions and many strokes, in addition to disabling peripheral artery disease. Progression of atheroma plaques has been associated with an increased risk of mortality. It is a challenge to determine in advance if atherosclerotic plaque will become unstable by calculating an index of instability. We proposed a score of ten parameters for the assessment of high-risk plaques, by quantification of 10 risk factors for acute cardiovascular events, most of them representing histological variables. (2) Methods: Carotid endarterectomy samples were collected from 10 highly symptomatic patients who matched for gender, age, risk factors, and plaque morphology. Samples were stained with hematoxylin-eosin, elastic van Gieson, Perls, and Mallory. Immunohistochemistry was performed using specific antibodies, such as CD31 for endothelial cells, CD68 for macrophages, and CD3 for T cells. (3) Results: For each plaque, the presence and/or number of histological features (fibrous cap thickness, lipid core size, plaque and cap infiltration with macrophages and lymphocytes, neovessels, intraplaque hemorrhage, parietal thrombi, and calcium deposits) were recorded on a simple semi-quantitative one- or two-grade scale. The study identified four stable plaques (SPs), two vulnerable plaques (VPs), and four unstable plaques (USPs). We found significantly more macrophages and neovessels in unstable plaques compared with stable plaques. The score for unstable plaques was higher than that for VP. (4) Conclusions: The study showed that symptomatic carotid disease is associated with an increased index score. The proposed algorithm for carotid plaque assessment may be useful for an imaging application.

**Keywords:** atherosclerosis; unstable plaque; vulnerable plaque

## 1. Introduction

Complicated atherosclerotic disease is the leading cause of morbidity and mortality [1]. Carotid atherosclerosis is also a known predictor of other cardiovascular (CV) events, such as stroke [2]. According to Virmani R et al., an unstable carotid plaque with an overlying surface disruption and consequent thrombosis is associated with a higher risk of early stroke [3].

In a previous study, we correlated histological features from resected carotid atherosclerotic plaques with atherosclerotic risk factors (RFs), but no prognostic score model is currently available for molecular imagistic application [4].

Carotid atherosclerotic disease requires a complex algorithm for the assessment of prognosis and treatment, using mainly histological variables [5]. The presence and degree of atherosclerosis, as defined by plaque presence detected in the carotid arterial system, have been used to estimate and classify an individual's stroke risk [6]. Various scientific papers underlie the distinction between stable (SP), vulnerable (VP), and unstable plaques (USP) [7,8]. In the literature, there is a lack of clarity in the analysis of the terms defining vulnerable and unstable atherosclerotic plaques. Many authors consider that they are similar terms [9,10], whereas other researchers consider that thin-cap fibro atheroma (TCFA) is a vulnerable plaque, and that atherosclerotic plaques having calcified nodules, erosions, and capsule rupture are unstable [11]. The nonstable plaque is more often used for symptomatic plaques due to its potential to lead to a vascular event (ischemic stroke, acute coronary syndrome, etc.).

Atherosclerosis (ATS) is an arterial disease characterized by the formation of atherosclerotic plaque in the intima of the vessel with secondary damage to the media. Inflammation and neovascularization have a significant role in plaque progression [12].

ATS is also a leading cause of mortality due to its complications. The most common complications of atherosclerotic plaques are ulceration, thrombosis, calcification, and intraplaque hematoma.

In our study, we applied a useful histologic classification [11] describing different ATS stages from stable to unstable plaques in patients undergoing carotid endarterectomy (CEA). The purpose of our study was to establish an instability plaque index, representing a score of ten risk factors, including eight histological factors. The proposed algorithm for carotid plaque assessment may be useful for an imaging application.

## 2. Materials and Methods

### 2.1. Patients

The study included ten selected patients (8 males and 2 females) aged over 60 years old (70.1 ± 7.7 years), who underwent carotid endarterectomy (CEA), from 1 January–31 December 2017. The selected cases had cardiovascular risk factors (hypertension, hyperlipidemia, diabetes mellitus, and smoking), and no symptomatic coronary artery disease. Patients were excluded if they had a history of surgery for carotid restenosis, severe pulmonary hypertension, or uncontrolled systemic hypertension.

### 2.2. Study Procedures

2.2.1. Carotid Endarterectomy

Carotid surgery was performed using a routine endarterectomy technique, and the plaque was removed with the minimum of trauma possible.

2.2.2. Risk Factor Analysis

The information on individual clinical properties (age, sex hypertension, smoking, history of the aortic disease, and previous inflammatory process) were obtained from medical records.

2.2.3. Histopathological and Immunohistochemical Analysis

Histological analysis of the aortic specimens was performed. Hematoxylin-eosin staining was used routinely as a standard diagnostic tool. Elastic Van Gieson's staining was used to analyze elastic fibers, Perls staining was used to assess evidence of old IPH, Mallory staining was used for fibrin, and von Kossa staining was used for calcium salts. Magnifications of $40\times$, $100\times$, $200\times$, and $400\times$ were used. A histological assessment was performed by an experienced pathologist using an optical microscope (CX41; Olympus

Corporation, Tokyo, Japan). The measurements were visualized using color image analysis software (QuickPHOTO MICRO 3.0, PROMICRA, S.r.o., Prague, Czech Republic).

Immunohistochemistry (IHC) was performed according to standard protocols on formalin-fixed, paraffin-embedded sections. IHC examination focused on inflammation and angiogenesis assessment using CD68 antibodies to stain macrophages, CD3 antibodies to stain lymphocytes, and CD31 for new vessels The endogenous peroxidase of deparaffinized sections was neutralized by hydrogen peroxide 3% for 10 min. Antigen unmasking was performed by incubating the slides in a pH 9 buffer at 98 °C for 45 min (Agilent, Santa Clara, CA, USA, catalog code K800421). For immunohistochemical detection, antibodies to CD68 (monoclonal antibodies Clone PG-M1; RTU, DAKO, Carpinteria, CA, USA, catalog code IS609), antibodies to CD3 (monoclonal antibodies Clone F7.2.38, DAKO, catalog code M725401-2), and antibodies to CD31 (monoclonal antibodies Clone JC70A, DAKO, catalog code IS610) were applied for 30 min followed by the streptavidin-horseradish peroxidase conjugate for 15 min. Peroxidase was developed using the DAB working solution (DAKO) and washed in deionized water. Sections were counterstained using hematoxylin Gill II (Sigma, Kawasaki, Japan). The negative control was obtained by the replacement of the primary antibody with PBS.

The assessment of the macrophages and microvessel density was performed by analyzing five histological fields at $200\times$ magnification for each case (one field corresponding to a manually traced area reported on 100,000 $\mu m^2$/0.1 $mm^2$). The results are expressed as percentages or mean values of the number of positive cells related to the studied area.

### 2.2.4. Morphometrical Analysis

For each plaque, the presence and/or amount of the histological features (fibrous cap thickness, lipid core size, plaque and cap infiltration with macrophages and lymphocytes, neovessels, intraplaque hemorrhage, parietal thrombi, and calcium deposits) were recorded on a simple semi-quantitative 1 or 2-grade scale.

### 2.2.5. Index Calculation

The plaques were divided according to the Modified American Heart Association (AHA) classification of advanced coronary atherosclerosis [11], allowing for differentiation of the 3 main types of ATS plaque—SP, VP, and USP—depending on their histological structure and their progression toward thrombosis.

The stable plaque is known to be a fibrous cap atheroma [7]. As Virmani noted, the plaque is stable when the fibrous cap is 2 mm thick [13].

The plaques that have a high risk of rupture are called vulnerable plaques [7,14]. The characteristic VP is called thin cap fibroatheroma (TCFA), which is considered the precursor of the ruptured plaque [15]. It has a thin fibrous cap infiltrated by macrophages and lymphocytes with rare smooth muscle cells (SMCs) and an underlying necrotic core [16].

According to Virmani et al. [11], there are 3 main types of unstable plaques: plaque erosion (PE), PR (plaque rupture), and calcified nodule (CN).

Plaque erosion is defined as the presence of thrombosis with the absence of an endothelial layer at the thrombosed sites [2]. There is no communication of the thrombus with the necrotic core. Thrombus is mostly mural and infrequently occlusive [11]. Nevertheless, eroded plaques do not have specific distinct morphological characteristics, making the identification of erosion-prone plaques a challenge.

Plaque rupture is the result of disruption of a thin fibrous cap that overlies a large necrotic core, causing the thrombogenic contents of the necrotic core to trigger thrombus formation. In contrast to eroded plaques, plaques undergoing rupture have distinct morphological characteristics, including a large necrotic core, a thin and inflamed fibrous cap thin (<200 μm thickness), and intimal vascularization, inflammatory cell infiltration, and spotty calcification [6,17,18].

Calcified nodules are characterized by thrombus formation over nodular calcification protruding into the lumen through a disrupted thin fibrous cap. These nodules are charac-

terized by an eruptive nodular calcification with underlying fibrocalcific plaque. Thrombus usually is non-occlusive [11].

The applied AHA morphological classification suggests that atherosclerotic changes depend on their histological structure and progression toward thrombosis [11]. The histological variables taken into consideration for plaque assessment are destabilized components of the plaque, including the thin fibrous cap, large lipid core, plaque inflammation and neovascularization, intraplaque hemorrhage, plaque calcification, and thrombi formation.

A fibrous cap thickness under $\leq 200$ µm (0.2 mm) indicates a high risk for plaque rupture [13,18]. A large lipid core having a size of more than 2 mm$^2$ indicates a high risk for plaque rupture [19]. The presence of inflammatory cells in atherosclerotic lesions is a sign of plaque growth and appears to play a role in the process of plaque progression [20]. Intraplaque neovascularization arises from newly formed microvessels that grow from adventitial vasa vasorum into the intima; they are characterized by leaky capillaries with an endothelial lining that is immature [21]. The neovascularization within carotid artery plaques is associated with neovessel rupture and intraplaque hemorrhage [22]. Plaque calcification represents the deposition of calcium salts within the atheromatous core, either in small spotty amounts or in large amounts as nodular deposits [23]. According to Lammie et al., the thrombus can develop on CN luminal surface as fibrin deposits and erythrocytes [24]. No plaque rupture was recorded in our study cases; we did not find any clear communication between the lipid core and the lumen because there was no break in the fibrous cap.

We took into consideration 10 risk factors present in the advanced ATS plaque, which can present different values: values with low risk or score 0 (lesional grade 1) or values with high risk or score 1 (lesional grade 2). By summing the scores of the risk factors, an index (sum of the scores) between 1 and 10 is obtained. The stable plaque has an index of 1–4; the vulnerable plaque has an index of 5; and the unstable plaque has an index of 6–10.

For example, case 1 was a woman (score 0) aged 75 (score 1), in whom the plaque presents the following lesional risk factors: FCT—948 µM (score 0), LC—9.3 mm$^2$ (score 1), number of CD68—52 (score 1), number of CD3—7 (score 0), number of CD31—38 (score 1), focal basal calcifications (score 0), IPH—absent (score 0), and PT—absent (score 0). The sum of the lesion scores yields an index of 4, which corresponds to the stable plaque.

Case 10 was a man (score 1) aged 61 (score 0), in whom the plaque presents the following lesional risk factors: FCT—15 µM (score 1), LC—2.05 mm$^2$ (score 1), number of CD68—39 (score 0), number of CD3—14 (score 1), number of CD31—24 (score 1), no calcifications (score 0), IPH—absent (score 0), and PT—present (score 1). The sum of the lesion scores yields an index of 6, which corresponds to the unstable plaque.

### 2.2.6. Statistical Analysis

The statistical analysis was performed using IBM SPSS Statistics 21 Software (IBM Corp, Armonk, NY, USA). The Mann–Whitney test was used for non-parametrical data analysis. Statistical significance was set at $p < 0.05$.

### 3. Results

The purpose of this study was to identify unstable carotid artery plaques at risk of rupture by calculating an instability index. We took into account several potential markers of instability; the definitions or assessments were based on widely accepted descriptions of unstable plaques.

Two grades of plaque instability were proposed (Table 1). Other clinical and biological risk factors present in the studied patients were not included in the assessment of the risk of instability. In patients' medical records, we found atherosclerotic risk factors: smoking (3/10 patients; 30%), hypertension (8/10 patients; 80%), dyslipidemia (5/10 patients, 50%), and diabetes (4/10 patients, 40%).

**Table 1.** Two grades of the proposed plaque instability index were determined by assessing clinical and histological risk factors.

| Risk Factors | Score | |
|---|---|---|
| | Grade 1 | Grade 2 |
| Age (years) | <70 | >70 |
| Sex | Female | Male |
| Fibrous Cap Thickness (FCT) | A band like FCT > 200 μm | <200 μm |
| Lipid core (LC) | Small LC < 2 mm² | Large LC > 2 mm² |
| CD68 | <50 cells | >50 cells |
| CD3 | <10 cells in cap | >10 cells in cap |
| CD31 | <20 per section | >20 per section |
| Calcium salts | Spotty only | Calcified nodules |
| IPH | Small amounts | Large amounts |
| Thrombi | Parietal | Occlusive |

FCT—Fibrous Cap Thickness, LC—Lipid core, IPH—intraplaque hemorrhage, CD68—macrophage, CD3—lymphocyte, CD31—vessels.

The most relevant histological differences between stable plaque and unstable plaque are shown in Figure 1 (thick FCT versus thin FCT) and in Figure 2.

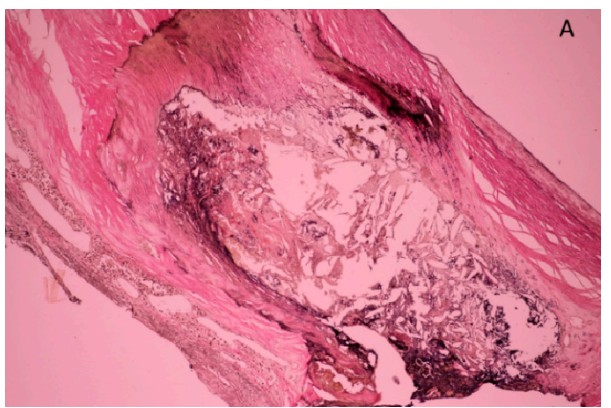
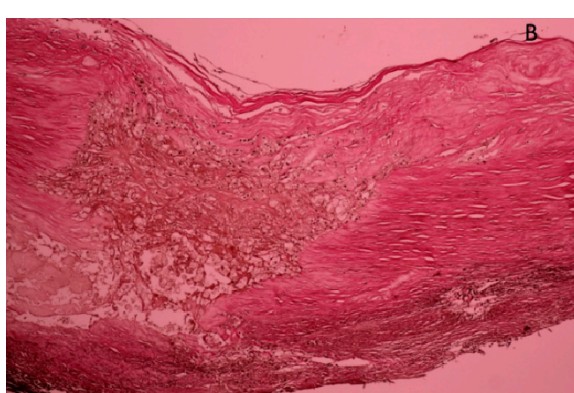

**Figure 1.** (**A**) Stable plaque (low instability score) has a thick FCT. (**B**) Vulnerable plaque (high instability score) has a thin FCT erosion (elastic Van Gieson staining of the elastic tissue; ×20 magnification).

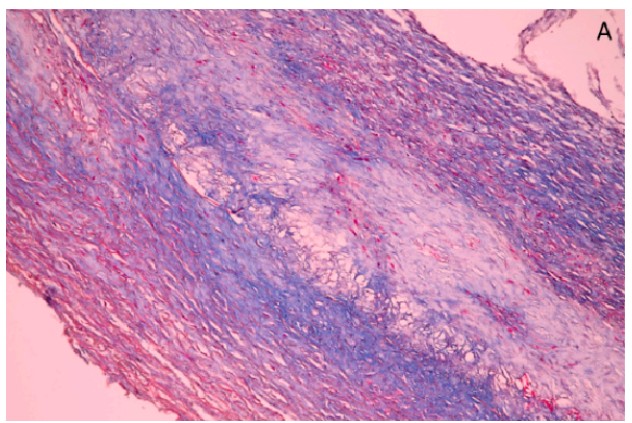
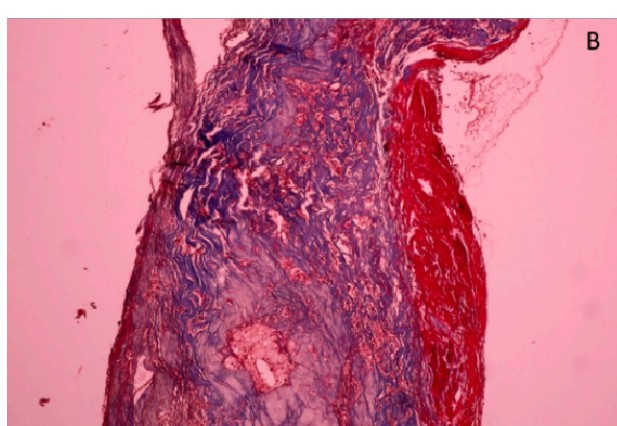

**Figure 2.** (**A**) Stable plaque (low instability score) has a thick FCT. (**B**) Unstable plaque (high instability score) has a parietal thrombus on endothelial erosion (elastic Van Gieson staining of the elastic tissue; ×20 magnification).

The selected patients (eight males and two females, aged over 60 years) who underwent CEA exhibited carotid stenosis produced by ATS plaques. By analyzing histological plaque features, we identified various ATS plaques, such as stable, vulnerable or TCFA, and unstable,

including erosive plaque and calcified nodules. The score assessment is shown in Table 2. The calculated score reflects the plaque progression: 1 to 4 for the four stable plaques (4/10 patients; 40%); a score of 5 for vulnerable plaques—plaque before thrombotic instability (2/10 patients; 20%); and 6 to 10 for the four unstable plaques (4/10 patients; 40%).

**Table 2.** The proposed plaque instability index in studied patients (*n* = 10).

| Instability Index (Score) | Patients | | | | | | | | | |
|---|---|---|---|---|---|---|---|---|---|---|
| | **1** | **2** | **3** | **4** | **5** | **6** | **7** | **8** | **9** | **10** |
| Sex | F (0) | F (0) | M (1) | M (1) | M (1) | M (1) | M (1) | M (1) | M (1) | M (1) |
| Age | 75 (1) | 67 (0) | 60 (0) | 60 (0) | 73 (1) | 63 (0) | 68 (0) | 71 (1) | 67 (0) | 61 (0) |
| FCT ($\mu$M) | 948 (0) | 1.115 (0) | 468 (0) | 552 (0) | 351 (0) | 7.7 (1) | 41.6 (1) | 2.45 (1) | 101 (1) | 15 (1) |
| LC (mm$^2$) | 9.3 (1) | 0.08 (0) | 0.94 (0) | 0.29 (0) | 0.62 (0) | 10.8 (1) | 9.34 (1) | 2.45 (1) | − (0) | 2.05 (1) |
| CD68 > 50 | 52 (1) | 88 (1) | 27 (0) | 31 (0) | 51 (1) | 19 (0) | 20 (0) | 67 (1) | 54 (1) | 39 (0) |
| CD3 > 10 | 7 (0) | 9 (0) | 5 (0) | 6 (0) | 14 (1) | 11 (1) | 12 (1) | 8 (0) | 6 (0) | 4 (1) |
| CD31 > 20 | 38 (1) | 20 (1) | 18 (0) | 14 (0) | 9 (0) | 28 (1) | 21 (1) | 6 (0) | 30 (1) | 24 (1) |
| Calcification | S (0) | S (0) | − (0) | − (0) | S (0) | − (0) | − (0) | − (0) | NC (1) | − (0) |
| IPH | − (0) | − (0) | + (1) | + (1) | + (1) | − (0) | − (0) | − (0) | − (0) | − (0) |
| PT | − (0) | − (0) | − (0) | − (0) | + (1) | − (0) | + (1) | − (0) | + (1) | + (1) |
| Diagnostic | SP | SP | SP | SP | EP | TCFA | EP | TCFA | NC | EP |
| Total score | 4 | 2 | 2 | 2 | 6 | 5 | 6 | 5 | 6 | 6 |

Instability index parameters were denoted by 0 (grade 1) or 1 (grade 2). F—Female, M—Male, FCT—Fibrous Cap Thickness, LC—Lipid core, IPH—intraplaque hemorrhage, PT—parietal thrombus, CD68—macrophage, CD3—lymphocyte, CD31—vessels, S—Spotty, deep calcium salts, NC = nodular calcification; SP = stable plaque, EP = erosive plaque, TCFA = thin fibrous cap atheroma.

We found significant differences in total index score (*p* = 0.01) between plaques with a high index score ($\leq$5) as compared to plaques with a low index score (<5) (Table 3).

**Table 3.** Clinical, histological, and immunohistochemical characteristics of carotid plaques.

| Variable | Plaques with Low Instability Index $\leq$ 5 (*n* = 4) | Plaques with High Instability Index > 5 (*n* = 6) | *p*-Value |
|---|---|---|---|
| Age > 70 years | 1 (25%) | 2 (33%) | 0.91 |
| Sex (Male) | 2 (50%) | 6 (100%) | 0.26 |
| FCT (<200 $\mu$m) | 0 (0%) | 5 (83%) | 0.04 |
| Lipid core (>2 mm$^2$) | 1 (25%) | 4 (67%) | 0.40 |
| CD68 (>50 cells) | 2 (50%) | 3 (50%) | 1.00 |
| CD3 (>10 cells in cap) | 0 (0%) | 4 (67%) | 0.11 |
| CD31 (>20 per section) | 2 (50%) | 4 (67%) | 0.76 |
| Calcium salts (calcified nodules) | 0 (0%) | 1 (17%) | 0.76 |
| IPH (large amounts) | 2 (50%) | 1 (17%) | 0.48 |
| Thrombi (occlusive) | 0 (0%) | 4 (67%) | 0.11 |
| Total index | | | 0.01 |

FCT—Fibrous Cap Thickness, IPH—intraplaque hemorrhage, CD68—macrophage, CD3—lymphocyte, CD31—vessels.

## 4. Discussion

According to Hafiane et al. [25], the intact thick fibrous cap (FC) is associated with a low risk of plaque rupture, whereas a thin FC is associated with a high risk of plaque break. Similarly, Nicolli et al. [26] noted that patients with acute coronary syndrome can

exhibit instability, including thrombus at the site of plaque erosion. The molecular imaging methods can be used to identify fissures of the FC and consequent thrombosis.

Our study emphasized the importance of histological plaque features in prognosis assessment. Our findings suggest that the size of fibrous cap thickness is a strong predictor of plaque instability [27]. In the analyzed samples, focal inflammation was a constant plaque feature, and previous reports found the macrophages were the main inflammatory cells [28]. The increased inflammatory state determines artery plaque instability [29,30]. Boer et al. noted that several types of inflammatory cells are identified in the carotid artery vulnerable plaque, and the presence of macrophages is significantly associated with the risk of plaque rupture [30]. In addition to macrophage accumulation, plaque fissures with superficial platelet aggregation are considered major criteria for plaque instability [31].

The state of chronic low-grade inflammation can contribute to the development of carotid plaque instability and stroke due to the activation of leucocytes. Inflammation promotes atherothrombotic complications by enhancing platelet reactivity and predisposing to plaque rupture and erosion.

Moreover, the atherosclerotic plaque infiltration with macrophages may enhance plaque angiogenesis [32]. Plaque angiogenesis is critical in the progression of atherosclerotic carotid plaque [22]. Similar to Fleiner et al. [33], we found a higher microvessel density in the carotid unstable plaques in association with macrophage infiltration compared with stable plaques, confirming that intraplaque neovascularization is also a marker for plaque instability.

According to other researchers [34], the symptomatic carotid disease has been associated with intraplaque hemorrhage and plaque instability. These features of the lesion can be influenced by the fragility of neovessels within the plaque [22]. Neovascularization developed from adventitial vasa vasorum as an adaptive response to hypoxia may cause subsequent intraplaque hemorrhage and plaque rupture [33]. Intraplaque hemorrhage is considered one of the identifying features of vulnerable plaque [34].

The true role of plaque calcification remains a matter of debate [6,35]. According to Nandalur et al. [23], micro-calcifications located within the fibrous cap are critical for plaque destabilization, whereas intraplaque macro-calcification has been connected with stronger stability of atherosclerotic plaques.

A digital quantitative method confirmed the features of unstable plaques obtained by semi-quantitative methods in specimens obtained by carotid endarterectomy, and was used to more precisely measured the amount of fibrous tissue, lipid core size, cap thickness, inflammatory cell infiltration, and intraplaque hemorrhage size as the major determining factors of plaque [36]. In addition, the digital method also measured CD68 and CD3 cells in the fibrous cap and plaque tissue separately, allowing for a more detailed characterization of each plaque.

The quantitative method was used to calculate a vulnerability index by histologically analyzing the carotid plaque: the smooth muscle cells (alpha-actin), lipids (Oil Red O), macrophages (CD68), hemorrhage (glycophorin A), and collagens [37]. The risk of future postoperative cardiovascular events was two times higher among patients having a high vulnerability index score than those having a lower score.

The central role of inflammation and proteases in plaque instability was highlighted by gene expression analyses in endarterectomies from patients with carotid stenosis. These studies identified the genes linked to plaque instability and also plaque phenotypes (i.e., calcification or lipid-rich necrotic core content) [38]. Perisic et al. proposed a panel of 30 genes, mostly transcription factors, to differentiate between plaques in symptomatic versus asymptomatic patients, including gene networks associated with atherosclerosis mapped to hypoxia, chemokines, calcification, the actin cytoskeleton, and the extracellular matrix [38]. In addition, Waden et al. performed an integrated analysis of symptomatic carotid stenosis patients at increased stroke risk with transcriptomic and clinical data and demonstrated that the state of plaque instability was more clearly related to enriched pathways coupled to neovascularization, ongoing coagulation, angiogenesis, iron homeostasis,

endothelial cell migration, and wound healing, which are all linked to neovessel formation and IPH [39].

There was moderate consistency between the assessment of plaque vulnerability by hematoxylin-eosin (HE) histopathological examination and by 2D-ultrasound ($p > 0.05$) [40]. However, there was a strong consistency between the assessment of plaque vulnerability by contrast-enhanced ultrasonography (CEUS) and histopathological HE ($p < 0.01$) [41] in patients who underwent carotid endarterectomy. The area under the curve (AUC) of the carotid plaque detected using CEUS was positively correlated with histologically quantitative parameters: microvessel density ($p < 0.001$), collagen type I/III ratio ($p < 0.001$), and MMP-9 and matrix metalloproteinase (MMP)-9 expression ($p < 0.001$) [42]. Ultrasound duplex scanning (USDS), the "gold standard" for the study of carotid artery atherosclerosis, can provide measurements such as the plaque area, the plaque volume index, and the area adjacent to the lumen. The ultrasound data were demonstrated by multivariate logistic regression analysis to be significant independent predictors of unstable plaques histologically verified by cap rupture with thrombosis, lipid core, plaque hemorrhage, neovascularization, calcification, and an increasing number of CD68+ and CD36+ cells (inflammatory markers) and CD31+ cells (neovasculogenesis markers) [43].

For unstable carotid plaques, demonstrated by a hypoechoic image of the carotid plaque on the contrast-enhanced ultrasound, the risk of rupture can be predicted by a high neutrophil-to-lymphocyte ratio (NLR) [44]. Waden et al. showed that a high Carotid Artery Risk (CAR) score is related to intra-plaque hemorrhage, angiogenesis, inflammation, and foam cell differentiation, and has a positive correlation with fibrinogen, white blood cell (WBC) count, and serum creatinine [39].

The vulnerability index calculated by image analysis in carotid plaques after staining to detect macrophages, lipids, plaque smooth muscle cells, and collagen fibers was used as a direct input in a Convolution Neural Network (CNN) algorithm, together with a two-dimensional ultrasound image of the corresponding intravascular pathological section (IVUS) [45]. In this study, the authors proposed a neural network-based method to determine the critical point of the vulnerability index that distinguishes vulnerable plaques from stable plaques. First, a vulnerability index is determined, above which the plaque is considered to be vulnerable plaque. According to the well-marked labeling, the convolutional neural network is used for classification, and then the vulnerable index points are continuously replaced. According to the fitting relationship between the vulnerability index and the accuracy rate, the vulnerability index with the highest classification accuracy can be found [45].

Therefore, our study provides a deeper insight into morphologic changes that characterize the unstable carotid plaque, and the proposed instability scores can be used in studies correlating histopathological assessment and ultrasound or clinical features of plaque instability.

*Limitations*

Our study had limitations, including the small group of patients with advanced aortic disease and no genetic studies. The small number of patients did not allow for more extensive statistical studies. The study reflects morphological particularities of aortic aneurysms in our geographic area. Additional study with a larger biomarker panel is required to determine the observed associations with other factors.

### 5. Conclusions

The plaque histology in patients undergoing CEA surgery is more complex than anticipated, and prognosis and treatment are thus challenging. Our study offers a model of an instability index for imaging application, allowing prognostic appreciations.

**Author Contributions:** Study design and concept: G.T. and D.B.; Performance of experiment: D.B., G.T. and M.E.; Data collection and analysis: D.B., M.E., B.G.I. and V.M.; Writing of manuscript: D.B., G.T. and V.M. All authors have read and agreed to the published version of the manuscript.



**Funding:** The authors received no financial support for the research, authorship, and/or publication of this article.

**Institutional Review Board Statement:** The study was conducted according to the guidelines of the Declaration of Helsinki, and approved by the Research Ethics Committee of the University of Medicine and Pharmacy from Iasi, Romania.

**Informed Consent Statement:** An informed consent form was signed by all the patients.

**Conflicts of Interest:** The authors declare no conflict of interest.

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
