# Peer review of "A Plaque Instability Index Calculated by Histological Marker Analysis of the Endarterectomy Carotid Artery"

_applsci, doi:10.3390/app12168040_

Round 1
Reviewer 1 Report
Review applsci 1767385
The article intituled “The plaque instability index calculated by histological markers analysis on endarterectomy carotid artery” present interesting results concerning the morphological characterization of atherosclerotic plaques and proposing and algorithm to classified then.
General comment
It is necessary to revise the English language, in order to improve the article comprehension.
E.g. line 27 sentence seen disconected to the text
In this paper, the process of plaque vulnerability and instability is debated.
Same occurring in line 50-51
line 185-186 please rephrase
Line 193 – please explain sentence, could be added in the discussion section
Line 220 please rephrase
In Material and methods:
line 72 (N males and N females) and mean age?
It would be interesting to include in the study if there were any exclusion factors to select the patients, and also includes, if possible, weight and BMI, and also cholesterol and triglycerides.
In line 75 the author comment that the patients presented (hypertension, hyperlipidemia, diabetes 75 mellitus, and smoking), but it would be interesting to know if the levels were similar, in a table. Because a patient can have 300 mg/dL cholesterol and another 150.
Page 101 pH 9 buffer, please add what buffer was used
In the result section it is important to show the histological results, at least the most representative imagens from the histological and imunohistological assays, since the article discuss the importance of the morphological analysis to determine plaque stability, it is important to the images showing the differences observed in the stable and unstable plaques, ad also it could be interesting to correlate the cardiovascular risks to the plaque characteristics, even though there are only 10 patients it could be discussed in the discussion section.
Author Response
We thank the Reviewer for their interest in our work and for helpful comments that will greatly improve the manuscript.
The article intituled “The plaque instability index calculated by histological markers analysis on endarterectomy carotid artery” present interesting results concerning the morphological characterization of atherosclerotic plaques and proposing and algorithm to classified then.
Answer. We have replaced the sentence.
General comment
It is necessary to revise the English language, in order to improve the article comprehension.
E.g. line 27 sentence seen disconected to the text
In this paper, the process of plaque vulnerability and instability is debated.
Same occurring in line 50-51
line 185-186 please rephrase
Line 193 – please explain the sentence, could be added in the discussion section
Line 220 please rephrase
Answers. We have rephrased or corrected the sentences.
In Material and methods:
line 72 (N males and N females) and mean age?
It would be interesting to include in the study if there were any exclusion factors to select the patients, and also includes, if possible, weight and BMI, and also cholesterol and triglycerides.
In line 75 the author comment that the patients presented (hypertension, hyperlipidemia, diabetes 75 mellitus, and smoking), but it would be interesting to know if the levels were similar, in a table. Because a patient can have 300 mg/dL cholesterol and another 150.
Answer. We added the following information about the risk factors in studied patients.
”In patients’ medical records we found as atherosclerotic risk factors: smoking (3/10 patients; 30%), hypertension (8/10 patients; 80%), dyslipidemia (5/10 patients, 50%), and diabetes (4/10 patients, 40%).”
Page 101 pH 9 buffer, please add what buffer was used
Answer. We used antigen retrieval solution (Agilent, catalog code K800421).
In the result section it is important to show the histological results, at least the most representative imagens from the histological and imunohistological assays, since the article discuss the importance of the morphological analysis to determine plaque stability, it is important to the images showing the differences observed in the stable and unstable plaques, ad also it could be interesting to correlate the cardiovascular risks to the plaque characteristics, even though there are only 10 patients it could be discussed in the discussion section.
Answers. In this article, we have proposed to present only the methodology for calculating the index.
Reviewer 2 Report
The idea of ​​the article looks attractive - the authors propose to use the index of non-stealth of the plaque in the carotid arteries.
However, while reading the manuscript, I had some remarks.
Major:
1. Since the study was conducted on patients, it is necessary to provide a clinical description of the individuals included in the study. For example, in the findings, there is a statement about symptomatic carotid atherosclerosis, but it is completely unclear what symptoms the patients had.
2. Some indicators in the proposed plaque instability index are non-specific or arbitrary values ​​are given (for CD68, CD3, CD 31). It is also unclear how the authors avoided subjectivity when evaluating a number of other indicators (intraplaque hemorrhage - Small amounts/Large amounts; Calcium salts - Spotty only/Calcified nodules).
3. The number of included patients is of fundamental importance; an additional set of patients is needed. If for the whole of 2017 only 10 patients were included, then nothing prevented the authors from continuing to include patients in the next 5 years.
4. I think the discussion section is poorly written. It does not contain information about what new knowledge was obtained by the authors. There is also a lot of fairly old data on some signs of plaque instability. At the same time, there is no information about studies using complex criteria for plaque instability similar to the approach used by the authors. For example, one could discuss the articles by Ignatyev IM et al. (doi: 10.1016/j.avsg.2020.08.145) or Cao Y et al. (doi: 10.1016/j.compmedimag.2020.101711.) etc.
Minor:
1. Some sentences contain incomplete information, for example "The study included ten selected patients (N males and N females) over 60 years old (mean age?)" - Lines 72-73; “The selected patients (N males and N females) and over 60 years old, undergoing CEA exhibited various ATS plaques” – Lines 185-186.
2. The phrase on lines 74-76 is not clear: "The selected cases operated for CABG surgery. The selected cases had cardiovascular risk factors (hypertension, hyperlipidemia, diabetes mellitus, and smoking), and no symptomatic coronary artery disease".
3. If the patients were subjected to CABG, then why did they not have symptoms of coronary artery disease? It should also indicate how many patients had cardiovascular risk factors.
4. At the end of the phrase "Patients were excluded if they had a history of surgery for carotid restenosis, severe pulmonary hypertension, and uncontrolled systemic hypertension," should there be a dot or some other information? (lines 76-78).
5. The following statement of the authors is surprising: "Informed Consent Statement: An informed consent form was signed by the school directors, parents, and students" (lines 247-248). Is this taken from another article?
References:
1. Ignatyev IM, Gafurov MR, Krivosheeva NV. Criteria for Carotid Atherosclerotic Plaque Instability. Ann Vasc Surg. 2021 Apr;72:340-349. doi: 10.1016/j.avsg.2020.08.145.
2. Cao Y, Xiao X, Liu Z, Yang M, Sun D, Guo W, Cui L, Zhang P. Detecting vulnerable plaque with vulnerability index based on convolutional neural networks. Comput Med Imaging Graph. 2020 Apr;81:101711. doi: 10.1016/j.compmedimag.2020.101711.
Author Response
We thank the Reviewers for their interest in our work and for helpful comments that will greatly improve the manuscript.
The idea of the article looks attractive - the authors propose to use the index of non-stealth of the plaque in the carotid arteries.
However, while reading the manuscript, I had some remarks.
Major:
- Since the study was conducted on patients, it is necessary to provide a clinical description of the individuals included in the study. For example, in the findings, there is a statement about symptomatic carotid atherosclerosis, but it is completely unclear what symptoms the patients had.
Answer. We have mentioned that the selected patients were symptomatic. We added information about atherosclerosis risk factors found in the studied patients.
- Some indicators in the proposed plaque instability index are non-specific or arbitrary values are given (for CD68, CD3, CD 31). It is also unclear how the authors avoided subjectivity when evaluating a number of other indicators (intraplaque hemorrhage - Small amounts/Large amounts; Calcium salts - Spotty only/Calcified nodules).
Answer. The considerations for which we chose the instability criteria were based on the central role of inflammation in plaque rupture and angiogenesis
- The number of included patients is of fundamental importance; an additional set of patients is needed. If for the whole of 2017 only 10 patients were included, then nothing prevented the authors from continuing to include patients in the next 5 years.
Answer. We have mentioned in the study limitations that the number of patients was reduced. We intend to continue our research.
- I think the discussion section is poorly written. It does not contain information about what new knowledge was obtained by the authors. There is also a lot of fairly old data on some signs of plaque instability. At the same time, there is no information about studies using complex criteria for plaque instability similar to the approach used by the authors. For example, one could discuss the articles by Ignatyev IM et al. (doi: 10.1016/j.avsg.2020.08.145) or Cao Y et al. (doi: 10.1016/j.compmedimag.2020.101711.) etc.
Answer. We thank the reviewer for his valuable help in improving our work. We have used the proposed interesting and important references, as well as other references to support the importance of knowing the morphological aspects of carotid artery plaque instability that can be used for the evaluation of ultrasound images.
Minor:
- Some sentences contain incomplete information, for example "The study included ten selected patients (N males and N females) over 60 years old (mean age?)" - Lines 72-73; “The selected patients (N males and N females) and over 60 years old, undergoing CEA exhibited various ATS plaques” – Lines 185-186.
Answer. Answer. We have added information and rephrased the sentences.
- The phrase on lines 74-76 is not clear: "The selected cases operated for CABG surgery. The selected cases had cardiovascular risk factors (hypertension, hyperlipidemia, diabetes mellitus, and smoking), and no symptomatic coronary artery disease".
Answer. We have rephrased the sentences.
- If the patients were subjected to CABG, then why did they not have symptoms of coronary artery disease? It should also indicate how many patients had cardiovascular risk factors.
Answer. We have removed that sentence that was introduced by mistake.
4. At the end of the phrase "Patients were excluded if they had a history of surgery for carotid restenosis, severe pulmonary hypertension, and uncontrolled systemic hypertension," should there be a dot or some other information? (lines 76-78).
Answer. We have removed that sentence that was introduced by mistake.
5. The following statement of the authors is surprising: "Informed Consent Statement: An informed consent form was signed by the school directors, parents, and students" (lines 247-248). Is this taken from another article?
Answer. We have corrected the statement.
Reviewer 3 Report
In this manuscript the authors show a histological index of instability of the atherosclerotic plaque after endarterectomy. The work is very interesting, even if the topic has already been extensively investigated in the literature and the results are rather expected.
Comments:
- Why didn't the authors also investigate oxidized LDL, recognized as one of the main markers of atherosclerotic plaque?
- Authors should better emphasize the impact and innovativeness of their study in the literature
Author Response
Reviewer 3
We thank the Reviewers for their interest in our work and for helpful comments that will greatly improve the manuscript.
In this manuscript, the authors show a histological index of instability of the atherosclerotic plaque after endarterectomy. The work is very interesting, even if the topic has already been extensively investigated in the literature and the results are rather expected.
Comments:
- Why didn't the authors also investigate oxidized LDL, recognized as one of the main markers of atherosclerotic plaque?
Answer. In this paper, we wanted to use morphometric and immunohistological analysis to highlight the presence of inflammatory cells as an effect of various factors of endothelial dysfunction.
- Authors should better emphasize the impact and innovativeness of their study in the literature
Answer. We added a phrase about the importance of our study:
”Therefore, our study provides a deeper insight into morphologic changes that characterize the unstable carotid plaque and the proposed instability scores could be used in correlation studies between histopathological assessment and ultrasound or clinical features of plaque instability.”
Round 2
Reviewer 1 Report
The authors responded most of my questioning. They comment that the article objective was to describe the methods used to create an index. My suggestion now was to …. Therefore, our study provides a deeper insight into morphologic changes that characterize the unstable carotid plaque and the proposed instability scores could be used in correlation studies between histopathological assessment and ultrasound or clinical features of plaque instability. Since the article did not describe anything about how these correlations can be done, the author could any some reference and suggestions about it.
And in the conclusions….
Our study offers a model of instability index for molecular imaging application, allowing prognostic appreciations.
I believe that the authors could change molecular imaging, to morphological imaging, since most of the characteristics were morphological
Little English changes:
Line 28 change early to in advance
line 204 the is a . in the middle of the sentence
Author Response
We thank the Reviwer for helpful comments that will greatly improve the manuscript.
The authors responded most of my questioning. They comment that the article objective was to describe the methods used to create an index. My suggestion now was to …. Therefore, our study provides a deeper insight into morphologic changes that characterize the unstable carotid plaque and the proposed instability scores could be used in correlation studies between histopathological assessment and ultrasound or clinical features of plaque instability. Since the article did not describe anything about how these correlations can be done, the author could any some reference and suggestions about it.
Answer: We agree with the reviewer that the relationship between histological and ultrasonographic or clinical aspects should be better highlighted. We have added examples and references.
And in the conclusions….
Our study offers a model of instability index for molecular imaging application, allowing prognostic appreciations.
I believe that the authors could change molecular imaging, to morphological imaging, since most of the characteristics were morphological
Answer: We have made the change.
Little English changes:
Line 28 change early to in advance
line 204 the is a . in the middle of the sentence
Answer: We have made the changes.
Reviewer 2 Report
I have read the authors' responses to my comments and those of other reviewers. It should be recognized that the authors have done a great job of improving the manuscript, especially in the Discussion section. After the correction of the manuscript, I have no comments.
Author Response
We thank the Reviewer for their positive comment.
Reviewer 3 Report
I have no further comments
Author Response

(The authors gave the same response as above.)
